# Heterogeneous Driving Factors of Carbon Emissions Embedded in China’s Export: An Application of the LASSO Model

**DOI:** 10.3390/ijerph181910423

**Published:** 2021-10-03

**Authors:** Jiajia Li, Yucong Liu, Houjian Li, Abbas Ali Chandio

**Affiliations:** College of Economics, Sichuan Agricultural University, Chengdu 611130, China; jiajia_li108@hotmail.com (J.L.); Yucong_liu1996@163.com (Y.L.); alichandio@sicau.edu.cn (A.A.C.)

**Keywords:** embodied carbon emissions, export, gender, female parliamentarians, China, LASSO model

## Abstract

With the steady growth of *CO_2_* emissions embedded in trade, the driving forces of emissions have attracted extensive attention. Most of the literature has verified a bundle of the influential factors; however, further analyses are necessary to understand the predominant and heterogeneous driving factors in different economies and/or industries. Accordingly, by applying the multiregional input–output (MRIO) model, this article firstly evaluates the embodied carbon emissions of China’s export from 1992 to 2020 in total volumes and by 14 industries. Then, the Least Absolute Shrinkage and Selection Operator (LASSO) estimations allow us to discover that urbanization, technology update and gross domestic product (GDP) are the leading three prioritizing factors in generating China’s export emissions. Interestingly, this paper discovers that raising the proportion of female parliamentarians contributes to an abatement of emissions. Furthermore, the empirical results suggest that the heterogeneities of those factors do exist among industries. For example, the percentage of females in parliaments turns out to have a larger effect among labor-intensive industries only. In facing with rapid globalization and economic development of China, this paper provides important policy implications towards specific industries in terms of mitigating trade emissions. It guides policy-makers to achieve “carbon neutrality” by avoiding carbon leakage in net-export countries such as China.

## 1. Introduction

The rapid growth of international trade has promoted the global demand for energy consumption, which in turn has contributed to greenhouse gas (GHG) emissions. Over the past two decades, about 25% of global carbon emissions generated from embodied emissions in export [1]. Hence, identifying the main driving factors of the trade emissions, particularly from the perspective of export, has crucial implications in terms of facilitating cooperation among countries to tackle climate change [2,3,4]. 

China, the largest global carbon emitter, aims to alleviate the dual pressures from international climate negotiations and domestic environmental degradation [5]. It has taken actions and set ambitious goals in regard to declining emissions, which puts forward higher requirements for carbon emission reduction [6]. For example, China has pledged to reach peak carbon dioxide emissions before 2030 and achieve carbon neutrality before 2060 [7]. In academia, a growing body of literature examined the trade carbon emissions in China [8,9].

China also generates the most carbon emissions embodied in its export in the world [4]. The evidence suggests that from 1995 to 2007, embodied carbon emissions in exports from the US, the UK and Japan accounted for 5–13%, 12–29%, and 6–22% of their total carbon emissions, respectively, while the percentage for China grew from 10% to 45% in the same period of time [10]. In 2013, approximately 29% of China’s carbon emissions were caused by the consumption of foreign customers [11], and hence, China is more likely to trap in the so-called “Pollution Haven” and even experience carbon leakage [12,13]. In order to tackle climate change, carbon reduction strategies should be implemented in China’s export by assessing its embedded carbon emissions and the driving forces in depth.

The driving factors of embedded carbon emissions in trade have been explored in the literature; see, for example, Islam et al. [2]; Kim and Tromp [14]; Deng and Xu [15], among others. Fewer studies realized the heterogeneity and priority of those driving factors embedded in trade among different regions and industries. In particular, China is a country with substantial industry differences in energy and economic structures [16], along with its large amounts of emissions in export. Thereby, it is particularly necessary to advance the understanding of export carbon emissions and their trend, and further, explore the priority and heterogeneity of the influencing factors both at the country and industry levels. On this basis, this article attempts to figure out the following research concerns: first, this article will evaluate the trends of the total and sub-industry embodied carbon emissions in China’s export over the past three decades. Second, it will examine the predominant driving factors of the total emissions in export and their heterogeneities across the industries and high-/low-carbon sectors accordingly. Based on the above results, policy designs are tailored to different industries and for different periods of time.

In practice, this paper estimates the embodied carbon emissions of export in total amount and by 14 industries from 1992 to 2020 in China by employing the multiregional input–output (MRIO) method. Then, we apply the Fully Modified Ordinary Least Squares (FMOLS) model as a benchmark regression estimation to examine the driving factors of the export carbon emissions at the country and industry level, respectively. Afterwards, we employ the LASSO model to identify the leading driving factors of the emissions and analyze the industrial heterogeneity of the driving factors. Additionally, following Fu and Zhang [17], we divide 14 industries into low- and high-carbon sectors (the mining, petroleum and coal products, chemical products, mineral products and metal products are regarded as a high-carbon sector, while the agriculture, food and tobacco products, textile, timber and furniture, rubber and plastic products, electronic equipment, machinery equipment, and transportation equipment belong to low-carbon sector). We also explore the predominant and heterogeneous driving factors of the emissions in export for the low- and high-carbon sectors.

The main contributions of this paper are mainly three-fold. First, this study covers 29-year data which are relatively longer in a period of time with the latest records. The dataset allows us to observe the trends of export emissions covering all the 14 industries, and then tailor the trade emissions mitigation strategies accordingly. Thereafter, it sheds light on policy designs in terms of forging a pathway for carbon neutrality. Second, this study innovatively adopts the LASSO model to explore the predominant and heterogeneous driving factors of China’s carbon emissions embedded in the export of total volume and by industry. Third, the findings of this article contribute to the literature in the aspect of verifying the environmental spillover effect of females, particularly focusing on the positive role of women’s political participation. According to the empirical results, females in parliament reduce the export emissions across the industries. It is worth noting that the coefficient of the variable turns out a larger effect in labor-intensive industries, which encourages policy-makers to pay extra attention to females in positions of advanced technology, which are promising to further mitigate the export emissions.

The remainder of the paper is organized as follows: Section 2 overviews literature. Section 3 describes the data in use, embedded carbon emissions calculation and shows the econometric models. The results and the relative discussions are reported in Section 4. Eventually, Section 5 summarizes the conclusions and provides policy implications. 

## 2. Literature Review

Embodied carbon emissions in trade have far-reaching implications in the division of regional emission reduction responsibilities [18]. Most of the earlier studies focused on developed countries; see, for example, Wyckoff [19]. Comparatively, studies on trade embedded carbon emissions in China and other developing countries have gradually received attention in recent decades due to the significant growing emissions in trade [20,21].

A large body of literature has explored the driving factors of embodied carbon emissions in trade. Economic growth is widely proved to be the most important factor of the growth in embodied carbon emissions [22]. Carbon intensity is another crucial factor [15]. In addition, a couple of studies suggested that urbanization stimulates the demand for production and consumption [23], which lead to extra embodied carbon emissions in export [9]. On the contrary, a larger amount of R&D expenditures decline the emissions in trade [24]. Furthermore, some other factors, such as trade openness [2], trade structure [14], foreign direct investment (FDI) [25] have been well discussed. In particular, some other studies investigated the effects of factors on the embodied carbon emissions of China’s export, such as population size, energy intensity, accession to the World Trade Organization (WTO) and educational level [4,9,26].

Besides the economic and demographic factors, social issues have received close attention. In particular, the literature stated that gender-related factors show environmentally friendly effects, and have potential effects on embodied carbon emissions. Existing evidence suggested that female politicians or CEOs tend to be more active in mitigating climate change, turning out to make environmentally friendly decisions [27,28]. For example, Ergas and York [27] proved that countries with the higher political status of women turn out lower *CO_2_* emissions per capita. Additionally, Li et al. [29]; Liobikiene et al. [30] confirmed that gender equality enhances environmental behaviors. The theoretical explanations for the above statements are, on the one hand, women have stronger pro-environmental behavior than men [31,32]; on the other hand, women are more vulnerable and become the main victims of climate change [33,34,35,36]. Furthermore, based on the theory of gender socialization, gender differences link to differences in values and social expectations. Notably, those differences are relevant to climate change actions which are more emphasized in females than in males [37,38]. Mavisakalyan and Tarverdi [39]; Wang et al. [40] used records of the proportion of females in parliament to verify the low-carbon role of women in determining carbon emissions. The two articles empirically discovered results of the low-carbon role of women in mitigating carbon emissions in trade among developed and developing economies.

It is worth noting that the driving factors of carbon emissions might exist priority, and are heterogeneous across regions and industries. For example, Bibi and Jamil [41] found that the impact of financial development on carbon emissions is positive in Europe, Central Asia, East Asia and the Pacific. On the other hand, this article verified that financial development reduced carbon emissions in Latin America and the Caribbean. Similarly, Li et al. [9] discovered that the technological decline embodied carbon emissions in trade for China, but had no effect on the emissions of Germany. In addition, due to different production materials and consumer demand, factors affecting carbon emissions in different industries might be also heterogeneous [42]. However, there is less literature to explore the heterogeneous factors in influencing carbon emissions among different industries, and examine the heterogeneities by applying models.

Regarding the methodologies of investigating the driving factors of the embodied emissions in trade, Liu et al. [43] applied the Log-Mean Divisia Index (LMDI) method, and decomposed the emissions into technology effect, structural effect and scale effect. Zhao et al. [3] used the structural decomposition analysis (SDA) to quantify the changes in the scale and structure of embodied carbon in China–US trade. The above-mentioned SDA and LMDI models are well suited in investigating the main drivers of the embodied carbon emissions in trade [44,45]. However, both of the two models have drawbacks in understanding the driving factors in terms of priority and heterogeneity. Specifically, the SDA model is likely to obtain non-uniqueness of decomposing results [46]. Moreover, the LMDI method shows a limited decomposition path [45]. Comparatively, the LASSO model is based on the thought of machine learning which has shown great potential for variable selection by the data instead of arbitrarily choose the factors, and it has been applied to the driving factors of carbon emissions [47,48]. Chen and Xu [49] also used the LASSO model to verify the main factors affecting carbon emissions of China. 

In sum, the aforementioned studies have extensively explored the various driving factors of carbon emissions embedded in trade and export across different countries and regions, which contributed to a comprehensive understanding of the causes of the growth of carbon emissions in international trade. However, the literature mainly focused on the driving factors of the trade embodied carbon emissions for country-level data. The analyses of the priority and/or heterogeneity of the driving factors in determining the carbon emissions embedded in traded at the industry-level of China are rare. Therefore, this article attempts to fill the gap.

## 3. Data and Methodology

### 3.1. Data

The carbon emissions embedded in export from 1992 to 2020 are calculated based on the following three data sources: multiregional input–output tables of China which were extracted from the World Input–Output Data (WIOD (Data access from: http://www.wiod.org/database/wiots16, accessed on 4 December 2019)), the export volumes by industry of China were collected from UNCOMTRADE (Data access from: https://comtrade.un.org/, accessed on 2 May 2021), and energy consumptions by different types for China were gathered from the Statistic Yearbooks of China (Data access from: http://www.stats.gov.cn/tjsj/ndsj/, accessed on 4 December 2019).

The other main independent variables include urbanization, R&D intensity, GDP per capita, the proportions of females in parliaments of China, which were collected from the World Bank (Data access from: https://data.worldbank.org/, accessed on 2 May 2021). First, Following Li et al. [9], we involve in the variable of urbanization in order to examine its effect on embodied carbon emissions in exports. Following the literature of Ouyang and Lin [50], economic growth leads to larger energy demand, which enhances carbon emissions; thus, we add GDP per capita as a proxy variable for the economic growth in the empirical model. On the contrary, the R&D intensity represents the technology level of a country in a certain period of time, and it contributes to the declining of the embodied emissions [24]. In addition, Mavisakalyan and Tarverdi [39] suggested that females in the parliaments are beneficial for the countries to promote climate change policies, which could reduce carbon dioxide emissions as well. 

Some other control variables are as follows: the dummy variables of China’s accession to the WTO (in the year 2001) and China’s proposal of the Belt and Road Initiative (in the year 2013). The two issues could stimulate international trade, and consequently, lead to the change of the embodied carbon emissions [9]. In addition, the exchange rate of RMB against the U.S. dollar is also included in the empirical models for the corresponding years. Fluctuation of the exchange rate will not only affect the marketing and production decisions of international enterprises, but also affect the investment of domestic and foreign investors, which will be associated with the embodied carbon emissions of China’s export [51]. Table 1 describes the main variables and shows the descriptive statistics of those variables.

### 3.2. Carbon Emission Calculation

First, this article calculates the direct carbon emission coefficients for 14 industries in China. According to the Intergovernmental Panel on Climate Change (IPCC) and the existing studies [9], eight major energy sources are extracted to form the carbon emissions. Afterwards, we can obtain the direct carbon emission coefficients matrix $R_{i,t}$.

Second, following Gao et al. [52], this article uses the input–output method to calculate the domestic direct consumption coefficient matrix *A^D^* according to Equation (1). The input–output method is an analytical technique that reflects the quantitative dependence of inputs and outputs among the parts of an economic system, which was researched and created by Wassily Leontief in the 1930s.
(1)$$X_{t}=(I-A_{i,t}{)}^{-1}Y_{t}$$
where $A_{i,t}$ is the direct consumption coefficient matrix, also known as the technology coefficient matrix, which is composed of the direct consumption coefficient $a_{ij}$ (the output value of the product or service of the *i*-th sector directly consumed by the total output of the *j*-th sector) and reflects the production links between the various product sectors of the national economy.$(I-A_{i,t}{)}^{-1}$ is the inverse Leontief matrix. $X_{t}$ represents the total social output column vector in year t. $Y_{t}$ is the social final product column vector containing other final products. *I* is the unit matrix.

Thereafter, we can calculate the carbon emission intensity $(F_{i,t}$) by multiplying the direct carbon emission coefficients matrix $R_{i,t}$ and the Leontief matrix $(I-A_{i,t}{)}^{-1}$ (see Equation (2)).
(2)$$F_{i,t\;}=R_{i,t}\times {\left(I-A_{i,t}\right)}^{-1}$$

The carbon emission intensity of 14 industries in China from 1992 to 2020 is obtained (see Table A1 in Appendix A), which refers to the sum of the direct and indirect carbon emissions coefficients of each industry. For the past 29 years, the embodied carbon emission intensity of all 14 industries has decreased. The declining trend suggests that China has made progress in energy saving and emission reduction. Industries with the lowest embodied carbon emission intensities include agriculture, food manufacturing and tobacco processing industry and textiles industries. The industry of petroleum and coal products experienced the greatest decline in carbon emission intensity, from 1.316 tons/million USD in 1992 to 0.394 tons/million USD in 2020.

Forth, by dividing the 14 industries into high-carbon and low-carbon sectors according to the threshold of the average carbon intensity of the industries, with those above the average value being high-carbon sector and those below being low-carbon sector [17]. Comparing with the literature, our obtained emission intensity is close to Ma et al. [53]. Furthermore, the low- and high-carbon sectors classifications of this article are almost the same as Fu and Zhang [17]. This evidence suggests the reliability of the carbon emission intensity in this article.

Finally, $C_{i,t}$ can be well calculated, which represents the embodied carbon emissions in export for industry *i* (see Equation (3)).
(3)$$C_{i,t}=F_{i,t}\times M_{i,t}=R_{i,t}\times (I-A^{D}{)}^{-1}\times M_{i,t}$$
where $F_{i,t}$ is the carbon emission intensity that was calculated in Equation (2). $M_{i,t}$ is the export volume of industry *i* for year *t*, which is downloaded from UNCOMTRADE.

### 3.3. Empirical Models

This section establishes a series of econometric models in terms of exploring in depth the driving factors, and their priority and heterogeneity, of China’s embodied carbon emissions in export by industry. 

We use the FMOLS model as a baseline regression model to empirically examine the driving factors of embodied carbon emissions from export of China (see Equations (4)–(6)). The FMOLS model, as refined by Pedroni [54], is able to correct for serial correlations and bias problems, which is appropriate for the date in use.
(4)$$CO_{2,t}=\alpha_{0}+\alpha_{1}GDP_{t}+\alpha_{2}Tech_{t}+\alpha_{3}Urban_{t}+\alpha_{4}Female_{t}+\alpha_{5}Rate_{t}+\alpha_{6}WTO_{t}+\alpha_{7}BRI_{t}+\varepsilon_{t}$$
(5)$$CO_{2,st}=\gamma_{0}+\gamma_{1}GDP_{t}+\gamma_{2}Tech_{t}+\gamma_{3}Urban_{t}+\gamma_{4}Female_{t}+\gamma_{5}Rate_{t}+\gamma_{6}WTO_{t}+\gamma_{7}BRI_{t}+\mu_{st}$$
(6)$$CO_{2,it}=\delta_{0}+\delta_{1}GDP_{t}+\delta_{2}Tech_{t}+\delta_{3}Urban_{t}+\delta_{4}Female_{t}+\delta_{5}Rate_{t}+\delta_{6}WTO_{t}+\delta_{7}BRI_{t}+\theta_{it}$$
where $CO_{2,t}$ represents the logarithm of total carbon emissions embedded in export of China in year t. $GDP_{t}$ denotes China’s gross domestic product per capita in year *t*. $Tech_{t}$ is technology update, represented by the number of R&D technicians per 100,000 people.$\;Urban_{t}$ is the level of urbanization. $Female_{t}$ denotes the proportions of females in parliaments of China in year *t*. $Rate_{t}$ symbolizes the exchange rate of RMB against the U.S. dollar in year *t*. $WTO_{t}$ is a dummy variable indicating whether China is a member of the World Trade Organization. $BRI_{t}$ is a dummy variable and it equals 1 after China implements the Belt and Road Initiative.$\;\alpha_{0}$ is the constant;$\;\varepsilon_{t}$ is the error term. Similarly, $CO_{2,st}$ in Equation (5) represents the logarithm of low-/high-carbon sectors’ carbon emissions embedded in export of China in year *t;*$\;\gamma_{0}$ is the constant; $\mu_{st}$ is the error term. $CO_{2,it}$ in Equation (6) represents the logarithm of embodied carbon embedded in export of industry *i* in year *t;*$\;\delta_{0}$ is the constant;$\;\theta_{it}\;$ is the error term. 

Then, we apply the LASSO model to further analyze those factors in determining the embedded emissions. The LASSO model was applied to time-series data, and this allows for efficient variable selection [55]. The reasons for using the LASSO model for this article are as follows. Generally, the LASSO model can solve the over fitting, multicollinearity problems and overcome the drawbacks of the general regression [56]. Second, it can identify the leading influential factors that affect the embedded carbon emissions and rank their importance according to the estimations and the data. Therefore, it provides greater flexibility of the models [47,57]. By applying the LASSO models, this article is capable of further analyzing the predominant factors and identifying industrial heterogeneities in determining the export embodied carbon emissions of China.

Particularly, the objective function of the LASSO model is to find the minimum number of independent variables, which is different from other traditional regression approaches (see Equation (7)). By setting the adjustment parameter  λ, we can achieve an acceptable level of information loss and retain only those variables which are the most valuable in the model [57]. When the distribution of the independent variables is unknown, the common methods for solving the adjustment parameters are the Cross-validation method and the Generalized Cross-validation method [47]. This article uses Cross-validation to determine the adjustment parameter *λ*. Thereafter, the coefficients β with little correlation are reduced or even compressed to zero.
(7)$$\beta =agr\;min\left\{\sum_{i=1}^{n}{\left(U_{i}-\sum_{j=1}^{m}\beta_{j}z_{ij}\right)}^{2}\right\},\;\sum_{j=1}^{m}\left|\beta_{j}\right|\leq \lambda$$

In Equation (7), *n* is the total number of the observations. $U_{i}$ is the dependent variable. $z_{ij}$ are the independent variables. *m* represents the number of the independent variables. *λ* is the adjustment parameter, and $\beta_{j}$ are the other parameters. Similarly, we estimate the LASSO model based on Equations (4)–(6).

## 4. Results 

### 4.1. Embedded Carbon Emissions in Export

Based on the subsection of 3.2, by applying the MRIO method to estimate the embedded carbon emissions in export, China’s total export volumes and their embodied carbon emissions from 1992 to 2020 are shown in Figure 1. The graph illustrates the specific trends of China’s exports and the embodied carbon emissions in the last three decades. Broadly speaking, it is clear that although the overall export volumes and their embodied carbon emissions were increasing in the past 29 years, the growing rate of the carbon emissions embedded in export was slower than that of China’s export volumes. This trend suggests that China made a progress towards low-carbon export. Specifically, from 1992 to 2001, the export volumes and embodied carbon emissions were increasing in parallel; however, since 2002, due to China’s accession to the WTO, the export volumes boosted significantly, and the embodied carbon emissions even sharply increased. Afterwards, the export volumes and embodied carbon emissions dropped significantly in 2009 partly due to the global financial crisis in 2008, and then maintained an upward trend until 2013. Since 2013, the exports and their embodied carbon emissions showed a slight fluctuation. Surprisingly, with the growing export volume during the period of 2019–2020, the carbon emissions embedded in export showed a slightly declining trend.

Table 2 shows China’s embodied carbon emissions generated from exports over the period of 1992–2020 by industry. The embodied carbon emissions are calculated according to Equations (1) and (2). In general, with the rapid growth of China’s exports, the embodied carbon emissions of various industries are rising over time. In particular, the largest increase is the electronic equipment industry (from 21.8 million tons in 1992 to 668 million tons in 2020), followed by machinery equipment, metal products and chemical products. On the contrary, the embodied carbon emissions in exports for the textile industry, petroleum and coal products and transportation equipment have decreased in recent years rather than increased consistently.

In 2020, electronic equipment, machinery equipment and basic metals are three industries with the most embodied carbon in exports. Thus, these industries are with larger pressures to reduce the carbon emissions embedded in export. In particular, the industry of the basic metals belongs to the high-carbon and resource-intensive sector. The export volume of the basic metals is not among the top three, but it has a high level of embodied carbon emissions due to its high carbon emission intensity. Therefore, policy-makers should pay particular attention to such industries and develop corresponding policies for embodied carbon reduction in exports.

### 4.2. The Baseline Results

Before the estimation regressions, we apply the Dickey–Fuller Generalized Least Squares (DF-GLS) test and the Phillips–Perron (PP) test to check the stationarity of the variables in the empirical models. All the estimation models pass the unit root test (see Table A2 in Appendix A). In addition, we use the Johansen–Juselius method [58] to analyze the cointegration relationship among the variables, and all the regression models pass the cointegration test as well (see Table A3 in Appendix A).

We first report the FMOLS results for China’s exports embedded in carbon emissions, low-carbon sector emissions and high-carbon sector emissions, respectively. The results of urbanization, R&D intensity, GDP per capita, the proportions of females in parliaments, the exchange rate of RMB against the U.S. dollar, China’s participation in WTO and its implementation of the Belt and Road Initiative on the impact of embodied carbon emissions in export are shown in Table 3.

For China’s total embodied carbon emissions in export, all the variables are significant in the FMOLS model, except for *WTO*. As expected, *Urban*, *GDP* and *BRI* are significantly positive with the total emissions. On the contrary, *Tech*, *Female* and *Rate* have negative significance, which suggests that an increase in these variables will release the total embodied carbon emissions from exports. 

For the results of the low-carbon and high-carbon sectors of China, the factors affecting embodied carbon emissions in export are mainly robust to the first column of total emissions. The only difference is that the exchange rate has an impact on the total carbon emissions, but not on the carbon emissions embedded in the export of the low-carbon and high-carbon sectors. It is worth noting that technology update plays a larger role in the high-carbon sector than in the low-carbon one. On the contrary, urbanization shows a greater impact on the low-carbon sector than the high-carbon sector. 

Owing to the differences in energy consumption and product demand among different industries, the factors that affect embodied carbon emissions may be different across the industries. Therefore, we establish the FMOLS regression estimations by industry. Table 4 presents the results, and it illustrates that the GDP per capita and China’s participation in the Belt and Road Initiative increase the carbon emissions embodied in most of the industries’ exports. By contrast, the improvement of the R&D intensity reduces the embodied carbon emissions among the industries, which is in line with the literature, see, Wang and Hu [24] for example. Furthermore, larger proportions of females in the parliaments promote a reduction in the embedded emissions across the 12 industries except for the following two industries: the mining industry and petroleum and coal products industry. 

Additionally, the level of urbanization has a significant positive effect on some particular industries, which is consistent with the result of the total embodied carbon emissions. However, it can be observed in Table 4 that the effect of urbanization on the emissions of agriculture, food and tobacco products, timber and furniture products, paper products, chemical products, mineral products, basic metals, electronic equipment and transportation equipment are not significant. Most of the industries with significant coefficients of *Urban* belong to the sector with high carbon emission intensity. The reason for this might be that the growth of the urbanization level stimulates the demand for high energy intensity products the most.

### 4.3. The LASSO Results

Table 5 shows the LASSO regression results for export carbon emissions reporting in the order of total emissions, low- and high-carbon sectors’ emissions. Generally, all the main explanatory variables have significant effects across the three regression models. 

Specifically, by ranking the coefficients of the explanatory variables, urbanization is the largest positive factor in affecting the total emissions, and it is followed by the variable of *Tech* and *GDP*. The urbanization process promotes carbon emissions through infrastructure establishment and the increase in residential energy consumption, which in turn stimulates the production and consumption of many other sectors. Consequently, urbanization significantly increases the embodied carbon emissions in export. The findings are aligned with the observations of Li et al. [9] and Ge et al. [23]. 

The coefficients of *GDP*, *WTO* and *BRI* are also positively significant, which confirm that the effects of these factors increase the embodied carbon emissions in export. On the contrary, the coefficients of the exchange rate and R&D intensity are negatively significant with the total emissions in export. In sum, the results of the LASSO model are mainly robust with the FMOLS model shown in Table 4.

Similar to the results of the total emissions, the coefficients for low-carbon and high-carbon sectors are in line with the expectations (see Columns (2) and (3) in Table 5). Again, Urbanization is the largest factor in affecting the embodied carbon emissions of the low-carbon sector. However, among the high-carbon sector, the R&D intensity takes the position of the largest effect on the embodied carbon emissions, which suggests that technology is more helpful in reducing the emissions for the high-carbon sector. Furthermore, although GDP per capita has a significant impact on embodied carbon emissions in total emissions and the high-carbon sector, it is less important in determining the embodied carbon emissions for the low-carbon sector. Therefore, heterogeneities do exist among different sectors. 

It is notable that the coefficients of *Female* are also significantly negative as the results in Table 4. The growth of female parliamentarians leads to stricter climate change policies [39]. Interestingly, although the proportion of females in parliaments is significantly negative in both high-carbon and low-carbon sectors, the role of women in the high-carbon sector is larger than that of the low-carbon sector. The reason for this heterogeneity could be that the high energy intensity industries also belong to the high-carbon sector. This sector has a higher potential to enhance emissions performance regarding pro-environmental policies with females’ political participation.

Comparing with the FMOLS regression results, the WTO accession turns out a significant positive effect on embodied carbon emissions across all the columns. Moreover, the exchange rate has a significant positive effect in both the low-carbon and high-carbon sectors.

Table 6 shows the estimation results of the LASSO model of the factors in determining the emissions by industry. Urbanization, GDP per capita and R&D intensity are the leading three prioritizing factors among the industries. Specifically, for those industries, including mining, timber and furniture products, petroleum and coal products, electronic equipment, machinery equipment and transportation equipment, urbanization is the most important factor in affecting the embodied carbon emissions. However, the coefficient of urbanization is not significant in agriculture, paper products, chemical products and basic metals. Figure 2 illustrates the ranking of coefficients for urbanization, GDP, technology and female among industries. Figure 2 suggests that urbanization contributes the most to the following three industries, namely, mining, machinery equipment and electronic equipment. Comparatively, the leading three industries, including chemical products, food and tobacco products and the Basic metals) are different when the driving force is *GDP*. Therefore, the heterogeneity of the driving factors in determining the carbon emissions embedded in export exists among the industries as well.

Furthermore, we find that the GDP per capita has no effect on the embodied carbon emissions of export in petroleum and coal products, electronic equipment, and machinery equipment. Additionally, Except for the petroleum and coal products, the coefficients of R&D intensity are negatively significant in all the other 13 industries, which indicates that technology update plays an important role in curbing embodied carbon emissions. In Figure 2, we observe the large impacts of R&D intensity on various industries, and the leading three industries are: mining, food and tobacco products, and textiles.

It is worth noting that the proportions of females in parliaments affects 14 industries with different extents, and it is the only factor that has a significant impact on embodied carbon emissions in all industries. The literature also generally supports the above results. Some scholars pointed out that females are vulnerable to the consequences of climate change [59,60]. Since females have greater concerns about climate change than males do [61], female representations guide countries to adopt more stringent climate change policies, which in turn mitigates the emissions. According to Figure 2, the larger proportions of female parliamentarians tend to encourage politicians to take environmentally friendly actions in agriculture, mining and textile industries the most, which is similar to the results of Wang et al. [40]. Comparatively, the role of females in parliaments show smaller effect in some technology-intensive industries. The results are robust compared with the coefficients of *Female* in low-carbon and high-carbon sectors.

Comparing with the FMOLS model results at the industry level, the amount of the industries with significance are more than the results of the LASSO model. For example, in the FMOLS regression results, urbanization is positively significant in only five industries, namely: mining, textile industry, petroleum and coal products, mineral products, and machinery equipment; while in the LASSO regression results, in addition to the above five industries, the urbanization is also positively significant in food and tobacco products, timber and furniture products, electronic equipment and transportation equipment. Similarly, in the FMOLS regression results, China’s accession to WTO is not significant among all the industries, while in the LASSO regression results, the variable is positive in all industries except for agriculture. In sum, the results of FMOLS and LASSO are generally robust, however, the LASSO models are capable to explore the predominant and heterogeneous driving factors among the industries. Therefore, the LASSO estimations are a priority to the basic FMOLS models in this article.

## 5. Conclusions and Policy Implications

In response to climate change, China has taken actions by participating in international climate treaties to mitigate carbon emissions, and has committed to peak *CO_2_* emissions by 2030 and achieve carbon neutrality by 2060 [7]. Embodied carbon emissions in export is a significant sector of generating carbon emissions for China [12]. It is therefore necessary to estimate the embodied carbon emissions for China’s export and further examine the predominant and heterogeneous driving factors at the country level and industry level, respectively. The analyses will not only guide for further low-carbon policies for international trade of China, but also provide specific evidence in terms of declining the emissions for other net export economies.

By using the MRIO method, this paper firstly calculates the total carbon emissions and 14 sub-industries from 1992 to 2020 embedded in China’s export. Then, we apply the FMOLS model as a baseline model to estimate the driving factors of embodied carbon emissions in export, and the models in use pass various robustness tests, including the DF-GLS test and the PP unit root test. Afterwards, by employing the LASSO model, the driving factors are ranked in terms of their importance. Notably, we examine the heterogeneity of the factors among the 14 industries. In addition, this paper classifies the 14 industries into low-carbon and high-carbon sectors and explores the priorities and heterogeneities in the driving factors for the two sectors as well. Several conclusions are drawn as follows.

This article discovers that the embodied carbon emissions in export show an upward trend from 1992–2020. Specifically, electronic equipment, machinery equipment and metal products are the three industries that increased the embodied emissions the most. The results of the FMOLS model and LASSO model are mainly robust. However, the results of the LASSO models take one step further by identifying that urbanization, technology update and GDP are the leading three prioritizing factors in influencing China’s export embodied carbon emissions in recent decades. Furthermore, heterogeneities do exist among industries. Specifically, urbanization explains the largest effect in the industries such as Mining and electronic equipment, whereas it does not show any significance in the agriculture industry. In addition, we find that the driving factors of embodied carbon emission also differ between low-carbon and high-carbon sectors. For example, the emissions of the high-carbon sector are largely influenced by technology updates, while urbanization has the most significance among the low-carbon ones. Interestingly, this paper discovers that raising the proportion of females in parliaments contributes to an abatement of the emissions, which turns out to have a larger effect among labor-intensive instead of technology-intensive industries.

The above results shed light on the following policy implications: 

(1) Industry differences should be emphasized in the process of reducing the embodied carbon emissions for China’s export. Energy intensity industries, such as petroleum and coal products, basic metal products and mineral products [62], should take corresponding measures to adjust their trade structure, reduce exports of energy-intensive products and shift to low-carbon intensive exports [15,52].

(2) Optimizing the energy structure and improving energy efficiency are crucial in terms of declining the emissions embedded in export. By using the MRIO method, this article finds out that although China’s carbon intensities across the industries have declined significantly in recent decades, there is still a gap compared with developed countries [63]. The government should strengthen technologies, particularly in the low-carbon sector, while reducing solid fuel consumption such as coal by transferring to clean energy use [64].

(3) Similar to Mavisakalyan and Tarverdi [39]; Wang et al. [40], our results show that women’s political empowerment contributes to the reduction in embodied carbon emissions. Therefore, it is necessary to involve women and let their voices be heard in political designs so as to address climate change. In addition, as women’s environmental spillover effect is larger in labor-intensity industries, governments should facilitate and encourage women to take high technology positions, and hence they have the potential to play a larger positive role so as to further reduce the emissions embedded in exports.

The limitations of this article are mainly the following two aspects: first, as the main focus of this paper is to estimate export carbon emissions and explore the driving factors, it does not forecast the carbon emissions embedded in export for the following decades, and hence this article cannot quantify the roadmap of achieving carbon neutrality in due course. Second, this paper applies the LASSO model based on macro-level data, further applications are in need to target the company-level data, and examine the heterogeneous driving factor to obtain more in-depth evidence.

## Figures and Tables

**Figure 1 ijerph-18-10423-f001:**
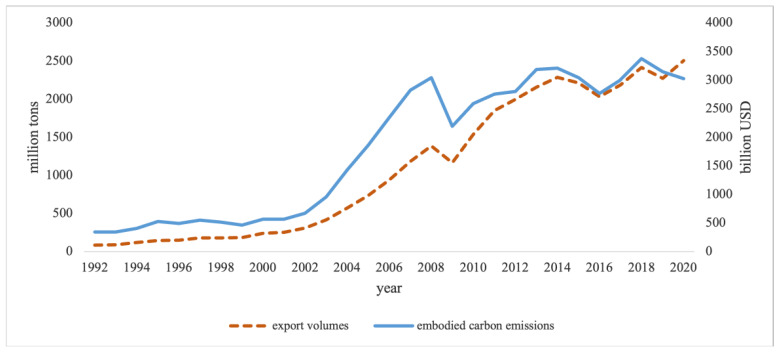
The trends of export volumes and the embodied carbon emissions for China from 1992 to 2020. Note: (1) The blue line, “export volumes”, stands for the total export volumes of China; (2) The red dotted line is the total embodied carbon emissions generated from the export.

**Figure 2 ijerph-18-10423-f002:**
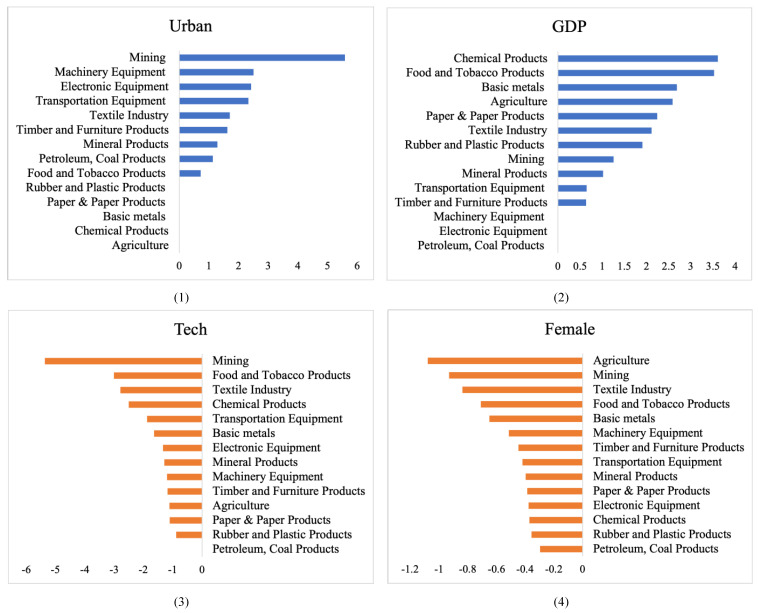
The ranking of coefficients for driving factors including (**1**) urbanization, (**2**) GDP, (**3**) technology update and (**4**) female parliamentarians across the industries. Note: The horizontal axis represents the coefficient of each driving factor.

**Table 1 ijerph-18-10423-t001:** Definition and descriptive statistics of the main variables.

Variable	Definition	Descriptive Statistics				
		Mean	Std. Dev	Min	Max	Obs.
$CO_{2}$	Logarithm of the total carbon emissions embedded in export of China (Abbreviated as “Total emissions” in the tables)	11.82	0.86	10.44	12.73	29
$CO_{2,s}$	Logarithm of the low-/high-carbon sectors’ carbon emissions embedded in export of China (Abbreviated as “Low-carbon” and “High-carbon” in the tables)	11.08	0.87	9.63	12.26	58
$CO_{2,i}$	Logarithm of the industrial carbon emissions embedded in export of China	8.56	1.45	5.43	11.19	406
WTO	China’s participation in the WTO (equals to 1 if t > 2001; 0 otherwise)	0.65	0.48	0	1	29
BRI	China’s implement the Belt and Road Initiative (equals to 1 if t > 2013; 0 otherwise)	0.28	0.45	0	1	29
Tech	R&D intensity (Number of technicians per 1 million people) of China	1.34	0.66	0.26	2.27	29
Rate	The exchange rate of RMB against the U.S. dollar	6.59	0.13	6.31	6.76	29
Urban	Logarithm of the urbanization of China	3.76	0.24	3.34	4.12	29
GDP	Logarithm of China’s GDP per capita	7.97	0.69	6.79	9.02	29
Female	Logarithm of the proportions of seats held by women in national parliaments of China	3.1	0.07	3.01	3.22	23

Note: The descriptive statistics for $CO_{2},$$CO_{2,s}$ and $CO_{2,i}$ are based on the calculations of the MRIO method which will be shown in Section 3.2.

**Table 2 ijerph-18-10423-t002:** The embodied carbon emissions in export by industry from 1992 to 2020.

Industry	1992	1996	2000	2004	2008	2012	2016	2020
Agriculture	18.3	20.1	17.4	26.1	22.1	24.0	28.0	25.2
Mining	4.7	6.1	4.3	7.0	12.2	7.3	5.9	8.0
Food and Tobacco Products	6.6	6.7	4.7	8.8	13.7	11.3	12.9	11.9
Textile Industry	88.1	108.8	116.1	213.5	362.7	286.5	291.5	264.1
Timber and Furniture Products	6.2	10.6	14.5	34.7	70.8	76.5	86.0	95.9
Paper and Paper Products	2.3	3.7	4.6	9.9	24.7	28.7	33.8	35.6
Petroleum, Coal Products	61.8	64.5	64.7	125.0	215.7	134.3	120.0	124.1
Chemical Products	7.0	14.0	19.1	42.6	96.0	119.4	124.7	158.2
Rubber and Plastic Products	8.5	14.3	16.7	40.2	88.3	99.8	110.0	120.6
Mineral Products	39.7	69.1	85.9	226.7	545.0	535.7	481.2	460.1
Basic metals	21.0	48.4	80.9	330.4	642.1	585.1	519.5	565.3
Electronic Equipment	52.4	70.4	43.3	95.7	242.8	234.7	250.4	388.2
Machinery Equipment	21.8	44.5	77.0	226.9	572.2	535.9	601.6	668.0
Transportation Equipment	6.4	10.6	17.8	42.2	130.6	120.7	97.5	100.1
sum	344.8	491.8	566.9	1429.8	3038.8	2800.1	2763.1	3025.2

Note: (1) The unit is million tons; (2) Due to the layout limitation, we only report eight-year data of the emissions to reflect the trend of the embodied carbon emission for various industries.

**Table 3 ijerph-18-10423-t003:** The FMOLS regression results for total emissions, low- and high-carbon sectors.

Variable	Total Emissions	Low-Carbon	High-Carbon
Urban	6.748 **	7.813 **	5.203 **
	(8.381)	(7.623)	(5.122)
Tech	−2.763 ***	−2.610 ***	−3.047 ***
	(0.658)	(0.599)	(0.795)
GDP	1.373 ***	0.899 *	2.113 ***
	(0.551)	(0.320)	(1.081)
Female	−6.536 ***	−6.102 ***	−6.338 ***
	(1.151)	(1.047)	(1.390)
Rate	−1.131 **	−1.148	−1.1257
	(0.932)	(0.848)	(1.125)
BRI	0.458 **	0.418 **	0.531 **
	(0.156)	(0.142)	(0.189)
WTO	0.129	0.153	0.354
	(0.177)	(0.161)	(0.213)
Cons	5.962 **	5.208 *	5.199 **
	(2.274)	(2.288)	(2.993)
Adj-R^2^	0.996	0.996	0.991

Note: (1) The numbers in parentheses are standard errors; (2) *, **, *** denotes significant at the 10%, 5%, and 1% levels, respectively.

**Table 4 ijerph-18-10423-t004:** The FMOLS regression results by industry.

Industry	Urban	Tech	GDP	Female	Rate	BRI	WTO	Cons	Adj-R^2^
Agriculture	−13.459	−0.421	5.538 **	−5.592 ***	2.849 ***	0.673 ***	0.178	12.672 *	0.706
	(6.311)	(0.496)	(1.921)	(0.867)	(0.702)	(0.118)	(0.133)	(6.230)	
Mining	5.546 **	−1.821	−2.336 **	2.213	−5.720 **	0.818 *	−0.0622	7.672 *	0.971
	(3.951)	(1.398)	(8.886)	(2.909)	(2.212)	(0.451)	(0.306)	(3.654)	
Food and Tobacco Products	1.988	−2.374 ***	2.078	−4.401 ***	−0.022	0.461 ***	0.195	6.735	0.967
	(5.580)	(0.438)	(1.699)	(0.767)	(0.620)	(0.104)	(0.118)	(5.509)	
Textile Industry	3.246 ***	−2.438 ***	1.652 **	0.518 ***	−1.002 *	0.5181 ***	0.136	12.044 *	0.987
	(6.579)	(0.516)	(2.002)	(0.903)	(0.731)	(0.122)	(0.138)	(6.495)	
Timber and Furniture Products	4.716	−2.033 ***	1.432	−6.075 ***	−0.938	0.582 ***	0.137	6.373	0.994
	(2.969)	(0.626)	(2.426)	(1.095)	(0.886)	(0.149)	(0.128)	(3.867)	
Paper and Paper Products	−1.612	−2.033 ***	3.677 *	−5.471 ***	−0.633	0.413 ***	0.155	7.050	0.997
	(7.140)	(0.561)	(2.173)	(0.981)	(0.794)	(0.133)	(0.151)	(7.049)	
Petroleum, Coal Products	2.876 **	−3.259 ***	−5.537	−2.092	−1.168	−0.025	−0.114	−11.703 **	0.973
	(1.171)	(0.877)	(3.400)	(1.534)	(1.242)	(0.208)	(0.235)	(5.028)	
Chemical Products	−1.711	−2.810 ***	4.291 *	−3.966 ***	−0.731	0.295 *	0.209	2.007 *	0.997
	(8.058)	(0.633)	(2.453)	(1.107)	(0.896)	(0.150)	(0.170)	(0.955)	
Rubber and Plastic Products	1.856	−1.773 **	3.610 *	−5.442 ***	−0.385	0.471 **	0.129	8.064	0.991
	(0.629)	(0.678)	(2.627)	(1.185)	(0.959)	(0.161)	(0.182)	(4.519)	
Mineral Products	1.908 **	−1.957 ***	2.303 *	−5.509 ***	−1.242	0.502 ***	0.173	10.348 *	0.995
	(0.486)	(0.588)	(2.279)	(1.028)	(0.832)	(0.140)	(0.158)	(7.390)	
Basic metals	−0.421	−3.402 ***	4.414 *	−9.378 ***	−1.242	0.854 ***	0.102	9.430 *	0.997
	(0.240)	(1.056)	(2.091)	(1.846)	(1.494)	(0.251)	(0.283)	(5.267)	
Electronic Equipment	14.231	−2.931 ***	−0.679	−6.253 ***	−1.864 *	0.380 **	0.084	−2.705	0.997
	(8.565)	(0.673)	(2.607)	(1.177)	(0.952)	(0.160)	(0.181)	(1.455)	
Machinery Equipment	5.46 *	−2.869 ***	−1.256	−7.606 ***	−1.483	0.37 **	0.253	1.175 **	0.891
	(2.492)	(0.667)	(0.585)	(1.167)	(0.944)	(0.158)	(0.179)	(0.384)	
Transportation Equipment	8.753	−3.247 ***	1.391 **	−6.421 ***	−2.261 **	0.137	0.039	3.400 *	0.992
	(3.900)	(0.620)	(0.405)	(1.085)	(0.878)	(0.147)	(0.167)	(7.799)	

Note: (1) The numbers in parentheses are standard errors; (2) *, **, *** denotes significant at the 10%, 5%, and 1% levels, respectively.

**Table 5 ijerph-18-10423-t005:** The LASSO regression results for total emissions, low- and high-carbon sectors.

Variable	Total Emissions	Low-Carbon	High-Carbon
Urban	2.756	2.996	2.047
Tech	−2.193	−1.936	−2.507
GDP	0.697	0.142	1.758
Female	−0.575	−0.528	−0.627
Rate	−0.279	−0.262	−0.289
BRI	0.181	0.151	0.229
WTO	0.032	0.040	0.012
Cons	0.236	0.281	0.191
Out-of-sample R^2^	0.936	0.942	0.920

**Table 6 ijerph-18-10423-t006:** The LASSO regression results by industry.

Industry	Urban	Tech	GDP	Female	Rate	BRI	WTO	Cons	Out-of-Sample R^2^
Agriculture		−1.117	2.585	−1.078	−0.725	0.592		−0.654	0.712
Mining	5.584	−5.362	1.255	−0.930		0.265	−0.051	−0.072	0.529
Food and Tobacco Products	0.725	−3.010	3.525	−0.709		0.312	0.145	−0.112	0.905
Textile Industry	1.703	−2.785	2.113	−0.836	−0.320	0.321	0.074	0.394	0.887
Timber and Furniture Products	1.626	−1.178	0.633	−0.447	−0.177	0.183	0.033	0.245	0.953
Paper and Paper Products		−1.102	2.242	−0.386	−0.142	0.125	0.030	0.121	0.967
Petroleum, Coal Products	1.134			−0.297	0.086		0.133	−0.123	0.662
Chemical Products		−2.502	3.610	−0.370	−0.241	0.132	0.060	0.049	0.922
Rubber and Plastic Products		−0.881	1.905	−0.355	−0.101	0.128	0.021	0.199	0.960
Mineral Products	1.284	−1.287	1.014	−0.397	−0.263	0.159	0.041	0.284	0.964
Basic metals		−1.637	2.681	−0.649	−0.194	0.242	0.015	0.317	0.914
Electronic Equipment	2.425	−1.332		−0.376	−0.198	0.093	0.027	0.287	0.945
Machinery Equipment	2.503	−1.198		−0.514	−0.097	0.138	0.106	0.333	0.940
Transportation Equipment	2.328	−1.877	0.647	−0.419	−0.345	0.03	−0.002	0.438	0.943

Note: The blank space suggests that the according variable is recognized by the LASSO model as a relatively unimportant factor for the corresponding industry, which means that the variable has no impact on this industry’s embodied carbon emissions in export.

## Data Availability

The dataset generated and/or analyzed during the present study is available from the corresponding author.

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
