# Peer review of "Heterogeneous Driving Factors of Carbon Emissions Embedded in China’s Export: An Application of the LASSO Model"

_ijerph, 2021, doi:10.3390/ijerph181910423_

Round 1
Reviewer 1 Report
Authors present an interesting research on the carbon emissions of China's export considering 14 different type of industries (sources) and some factors as driving force of the increasing/decrasing of carbon (or GHG) emissions.
In the discussion section, in order to privide a more complete dissertation in this topic, it could be interesting add some specification regarding: carbon emission - export trade trend - healthy city development. Furthermore, the concept of healthy city could be related to the urbanization rate (considering one of the most relevant driving force for the increasing of carbon emission in export trend). Do authors consider the global urbanization rate or a local one? please specify.
The paper is structured and written in a satisfactory way. I suggest to improve the fluidity of some sentences such as in line 166-172 with a proof reading. It is also necessary clarify all the acronnyms. Moreover, releted to Figure 1, please add some comments about the intersection of "trade volume" and "embodied carbon emissions". Finally, think to insert the industry list in the introduction section and to specify which of them are low-carbon and high-carbon productors right away.
Author Response
Please see the reply in the attachment. Thanks!

Reviewer 2 Report
The conducted research is quite interesting. Nevertheless, the article must also be assessed in terms of its content. The authors in the Introduction presented an introduction to the subject. They also did a very brief literature review. This part is deficient. I would propose a research hypothesis. Alternatively, research questions can be asked.
The layout of the work is not entirely correct. Section 2 Literature review may be at this place. Alternatively, this section may enrich the scientific discussion in part after the research results in the Discussion section. Line 150-157 must be moved to the Introduction section immediately before the sectioning information.
The text on pages 8 and 9 relating to Table 3 and Figure 1 should be moved to the beginning of the Results section. These are research results.
The Discussion section is missing. I understand a discussion as referring to other studies after presenting my research results. There was some discussion in section 2 for a comparison of the results achieved by other authors. Alternatively, part of the text can be moved from section 2 to the new Discussion section. In my opinion, doing research without a clear comparison and reference to other research results in the fact that the obtained results cannot be properly assessed. The results can also be presented together with the discussion.
The Conclusions section is too long. It needs to be shortened. You can focus on the main results. Which types of industries should be developed because they have low carbon dioxide emissions, and which ones should be reduced because they emit a lot. The conclusions should be synthetic. There is no need to re-write about the research assumptions, but to provide the results.
The greatest doubts arise from the use of the proportions of females in parliaments as the variable. In my opinion, this has no substantive justification. I am also not convinced by the positions to which the authors refer. This is a factually unjustified parameter. It is not related to the achieved results. In general, carbon dioxide emissions are systematically declining due to changes in production technology and world political pression. The greater proportion of women in parliament, on the other hand, results from trends in politics. The aim is for both sexes to be equal. Of course, the statistical results show a correlation. In terms of content, however, these parameters cannot be combined with each other. You might as well find some other absurd parameter that correlates. So please remove from the article text and research results which concerns the proportions of females in parliaments.
Tables are too extensive. You can reduce the font and place it in the text on a vertical page layout. Alternatively, in the case of very large tables, they can be placed at the end of the article, in the appendix.
There are mistakes in the text. The numbering of lines is wrong because you start over a few times. This is probably due to the different layout of the pages (landscape or portrait).
The article is too long. It has 27 pages. Part of this is due to blank pages and landscape pages. You also need to remove unnecessary text, redundant descriptions and repetitions in the results section.
The literature for the article covers 80-90% of the positions of Chinese authors. The research results were not compared with the results of other scientists obtained in other countries. Therefore, the review of the literature and the scientific discussion should be extended.
Author Response

(The authors gave the same response as above.)

Round 2
Reviewer 2 Report
The authors put a lot of work into improving the article. It looks better now. It is largely acceptable. However, I still lack a very substantive justification why the greater participation of women in parliament contributes to good climate change. Authors need to justify more on the basis of social literature why this is so. What characteristics of women affect it. This is crucial.
Author Response
Dear referee,
The reply is in the attachment. Thank you for checking it.
Best regards,
Jiajia
